# Low-Cost UVBot Using SLAM to Mitigate the Spread of Noroviruses in Occupational Spaces

**DOI:** 10.3390/s22228926

**Published:** 2022-11-18

**Authors:** Fanxin Wang, Harris Junaid Nisar, Yao Li, Elbashir Araud, Thanh H. Nguyen, Thenkurussi Kesavadas

**Affiliations:** 1Health Care Engineering Systems Center, Department of Mechanical Science and Engineering, University of Illinois at Urbana-Champaign, Champaign, IL 61801, USA; 2Health Care Engineering Systems Center, University of Illinois at Urbana-Champaign, Champaign, IL 61801, USA; 3Department of Mathematics, Harbin Institute of Technology, Haerbin 150001, China; 4Holonyak Micro & Nanotechnology Laboratory, University of Illinois at Urbana-Champaign, Champaign, IL 61801, USA; 5Institute of Genomic Biology, University of Illinois at Urbana-Champaign, Champaign, IL 61801, USA; 6Research and Economic Development, University at Albany—State University of New York, Albany, NY 12222, USA

**Keywords:** robot disinfection, UV light, norovirus, SLAM, autonomous robot

## Abstract

Noroviruses (NoVs) cause over 90% of non-bacterial gastroenteritis outbreaks in adults and children in developed countries. Therefore, there is a need for approaches to mitigate the transmission of noroviruses in workplaces to reduce their substantial health burden. We developed and validated a low-cost, autonomous robot called the UVBot to disinfect occupational spaces using ultraviolet (UV) lamps. The total cost of the UVBOT is less than USD 1000, which is much lower than existing commercial robots that cost as much as USD 35,000. The user-friendly desktop application allows users to control the robot remotely, check the disinfection map, and add virtual walls to the map. A 2D LiDAR and a simultaneous localization and mapping (SLAM) algorithm was used to generate a map of the space being disinfected. Tulane virus (TV), a human norovirus surrogate, was used to validate the UVBot’s effectiveness. TV was deposited on a painted drywall and exposed to UV radiation at different doses. A 3-log (99.9%) reduction of TV infectivity was achieved at a UV dose of 45 mJ/cm^2^. We further calculated the sanitizing speed as 3.5 cm/s and the efficient sanitizing distance reached up to 40 cm from the UV bulb. The design, software, and environment test data are available to the public so that any organization with minimal engineering capabilities can reproduce the UVBot system.

## 1. Introduction

There is a need for approaches to mitigate the transmission of pathogenic viruses among workers, patients, students, and customers when they are in close proximity in a workplace. For instance, studies have shown that workers in diverse occupations, such as the meat industry, the health care sector, and first responders, have expressed a high prevalence rate of COVID-19 during the pandemic [1,2]. In another example, over 260 health care workers at a Maryland hospital developed norovirus (NoV) infection symptoms during a NoV outbreak; this number was 300% higher than the number of infected patients [3]. About 25 health care personnel at a long-term care facility in Oregon expressed NoV symptoms during another NoV outbreak [4]. Therefore, disinfection of occupational areas is highly recommended to slow down or eliminate the spread of pathogens among workers. Norovirus (NoV) is a very contagious pathogen and can persist on different types of fomites and surfaces for long periods of time [5]. It has been shown that a few particles of NoV can cause infection [6,7,8]. The CDC estimated that about 23 million illnesses are attributed to noroviruses each year in the US. Workers in the health care sector, cruise ships, nursing homes, hotels, retirement centers, and schools are highly susceptible to infection during NoV outbreaks [9,10,11]. NoV can infect adults and children, and the symptoms include severe diarrhea, vomiting, nausea, and stomachache [7,8]. The virus can be transmitted through person-to-person contact, consuming contaminated food, or touching contaminated surfaces [12,13].

Norovirus belongs to the caliciviridae family. It is a nonenveloped virus with a (+)ssRNA genome encapsulated in an icosahedral virion 32–35 nm in size [14]. There is no robust cell culture system to study NoV infectivity; therefore, surrogates such as Tulane virus (TV) are used to study NoV disinfection [14,15]. TV has been used as a NoV surrogate because it utilizes the same receptor that is used by NoV to attach to the host cell (histo-blood group antigens). TV and NoV are also both enteric viruses and have the same capsid and genome structure [16].

UV irradiation is widely known for its disinfection ability to inactivate pathogenic viruses and bacteria [17,18,19]. It also has many advantages over chemical disinfection methods. For instance, it does not generate residual chemicals, requires minimal labor, and is easy to maintain and operate [20,21]. Currently, there are several commercial UV disinfection robots for indoor usage, such as the UVD Robot [22], LightStrike Germ-Zapping Robots by Xenex Disinfection Services [23], TMiRob [24], Tru-D 59SmartUVC [25], UVC Robot by UVCLight [26], Decimator by Addverb Technology [27], SmartGuardUV disinfection robot by Fetch [28], and Lavender by Geek+ [29]. The UVD Robot has a 360-degree disinfection coverage and is capable of 2.5 h of operating its UV module. However, the weight of 140 kg and the battery charging time of 6 h are strict restrictions on the mobility of the robot. The high retail price of USD 85,000 also limits its use to large and essential facilities (such as hospitals). LightStrike Robot makes use of high-power UV light to disinfect a room in 2 min. The robot does not integrate any navigation function and sells for USD 94,000. Tru-D SmartUVC does not have any navigation function either and costs USD 90,000. The UVC Robot can work on disinfection for 4 h after 4 h of charge. It features autonomous disinfection on preset routes and has a price of USD 35,000. The Decimator, Lavender, and Smart Guard UV integrate 2D LiDAR and a 3D depth camera. The Decimator can operate for up to 4 h after 30 min of charging. Smart Guard UV can operate 8 h after 4 h of charging, and the price is as high as USD 157,000.

The objective of this study was to design and test a low-cost (less than USD 1000, Appendix A
Table A1), lightweight, and intelligent autonomous UV light-based disinfection robot (UVBot) to reduce, and potentially eliminate, the need for human labor to mitigate virus transmission in indoor environments. Our work has four major contributions: (1) We tested the disinfection ability of commercially available UV lights with real virus experiments. (2) We designed a low-cost, UV light-based disinfection robot. (3) We developed software that allows users to control the UVBot to perform basic disinfection tasks and review the disinfection map. The default settings of the UVBot promise the inactivation of at least 99.9% of single-stranded RNA viruses. (4) We tested the UVBot in two indoor environments (corridor and office) to demonstrate its autonomous disinfection process.

## 2. Materials and Methods

### 2.1. Sanitizing Efficiency Experiment

#### 2.1.1. Cell Culture and Virus Propagation

Due to the lack of robust cell culture to grow NoV and study its disinfection, we selected Tulane virus (TV) as a model to determine the suitable inactivation time and UV dose that can fulfill the FDA regulations of 3-log viral reduction (99.9% disinfection). TV is a nonenveloped virus that has a virion size of 35 nm and a positive-sense single-strand RNA genome (~6.7 kb) [8,10]. It has been reported that viruses with longer single-stranded RNA have higher inactivation rates compared to those with shorter genomes [18,30,31]. Thus, TV is expected to be a more suitable model to study NoV disinfection and more resistant to 254 nm UV irradiation than most other airborne viruses, such as SARS-CoV2 and the influenza virus. The rhesus monkey epithelial cell line (MA104) was obtained from the American Type Culture Collection (ATCC in Manassas, VA, USA) and the TV was a generous gift from Xi Jiang, Cincinnati Children’s Hospital. The TV was propagated in MA104 cells in Dulbecco’s Modified Eagle’s Medium (EMEM) supplemented with 10% heat-inactivated fetal bovine serum (FBS) (Invitrogen), as previously described [17]. In short, a T175 flask of MA104 cells at 90% confluence was inoculated with TV at a multiplicity of infection (MOI) of 0.1. The infected cells were incubated at 37 °C in a 5% CO_2_ environment for 1 h, with hand agitation every 10 to 15 min to facilitate the virus attachment. After 1 h of incubation, 18 mL of EMEM containing 2% FBS was added to the flask and incubated at 37 °C and 5% CO_2_ until 80% of the cells showed a cytopathic effect (CPE). The virus was harvested by three cycles of freeze and thaw. The harvested virus was stored at −80 °C.

#### 2.1.2. Virus Titer Quantification

TV titers were quantified using a standard plaque assay [14]. Briefly, a standard plaque assay was conducted in MA104 cells seeded in 6-well plates for 24 h at 37 °C in a 5% CO_2_ environment. UV-irradiated samples and control samples were diluted in 10-fold serial dilutions in EMEM supplemented with 2% FBS. The diluted samples (400 µL) were added to each well of MA104 cells. The infected cells were incubated at 37 °C in a 5% CO_2_ for 1 h with hand agitation every 10–15 min to allow for virus attachment. After 1 h of incubation, 2.5 mL of overlay solution containing 2X MEM and 1% low-melting-point agarose was added to each well. The 6-well plates were incubated at 37 °C in a 5% CO_2_ atmosphere for 48 h to allow them to solidify. After the 48-h incubation, the cells were fixed with 10% formaldehyde and stained with 0.05% crystal violet dissolved in 10% ethanol.

#### 2.1.3. UV Disinfection Experiments

To mimic the real scenario of virus contamination of classrooms, patient rooms, and other spaces, all the UV-exposure experiments were conducted on a piece of drywall that was primed and painted with a commercial paint (BEHR ULTRA, Santa Ann, CA, USA). All virus inactivation experiments were conducted under a commercial UV bulb (USHO, GPH843T5L/4P. Hg 19261-L109, 1000Bulbs, Mesquite, TX, USA). The UV spectrum was measured using a BLK-C-50 spectrometer (StellarNet Inc., Tampa, FL, USA). The UV intensity was measured using a radiometer/photometer (International Light, Model IL1400A, Peabody, MA, USA ) with a QNDS2 # 24710 detector ((International Light, Model IL1400A, Peabody, MA, USA). Briefly, 100 µL of TV was suspended in an artificially made saliva (0.13 g of CaCl2∙H2O, 0.42 g of NaHCO_3_, 0.11 g of NH_4_Cl, 0.88 g of NaCl, 1.04 g of KCl, and 3.00 g of porcine mucin dissolved in 1000 mL of distilled water) at a ratio of 1:1 and placed on the drywall under the UV source. The virus droplets left on the drywall were exposed to UV at different distances from the UV source (10, 20, 30, 40, and 50 cm) for various intervals of time (10, 20, 30, and 40 s). The UV doses ranged from 1 to 176 mJ/cm^2^. The exposed virus droplets were collected after each treatment. The initial virus titer for each experiment was ~2 × 104 PFU/mL. All experiments were conducted in a biosafety cabinet at room temperature. The surviving TV was quantified using a standard plaque assay. The negative control was the virus suspension deposited on the drywall and covered with a dark petri dish to prevent its UV exposure.

#### 2.1.4. Statistical Analysis

All experiments were conducted in three independent replicates. The survival of the viruses was assessed by a virus infectivity assay with the standard plaque assay and plotted as log_10_(N_0_/N) as a function of the UV dose or the exposure time, where N is the virus infectivity after a given UV dose or time, and N_0_ is the virus infectivity before the virus exposure to the UV irradiation. The UV dose was calculated as described previously [17,18] and expressed as the product of UV Intensity (mW/cm^2^) and Exposure Time (seconds) in mJ/cm^2^. Linear regression was used to study the inactivation kinetics. To compare the inactivation rate of different UV doses, we applied analysis of covariance (ANOCOVA) in OriginPro to compare the TV inactivation rate at different UV doses and exposure times [19]. A *p* value of (<0.05) was used to determine the level of significance.

### 2.2. UVBot Mechanical Design

The UVBot is designed to hold up to four 36-inch UV lamps, a power supply, sensors, and a microcontroller. We chose the iRobot Create 2 mobile robot as the base for our design because of its low cost of USD 200 and 9 kg weight capacity. A base structure and lamp holders were modeled and 3D printed in polylactic acid (PLA) using a standard fused deposition modeling (FDM) printer. The base was designed to easily fit and mount on the iRobot Create while leaving access to the buttons on the robot. The base structure was kept separate from the lamp holders so that the number of lamps could be adjusted.

Four different ring structures were designed and laser-cut out of acrylic to hold the various components and to offer support. The overall height of this structure supports the lamps and holds all the electrical components. The ring structure allows for easy wire pass-through and wire management. The overall assembly and dimensions of the UVBot can be seen in Figure 1. Figure 2 summarizes all the key mechanical components and how they were put together.

### 2.3. UVBot Electrical Design

Our implementation can be separated into two main functionalities: autonomous movement and sanitization. We discuss each of these separately, however, they come together to allow for autonomous disinfection. The entire system is summarized in Figure 3.

#### 2.3.1. Sanitization

To disinfect and sanitize surfaces, we used high-powered UV lamps. In our setup, we used USHIO 3000343 G36T5L/4P Germicidal (1000Bulbs, Mesquite, TX, USA) lamps, which are low-pressure mercury arc lamps that emit radiation peaking at 253.7 nm (UV-C). This lamp was chosen because of its height (84.6 cm) and its disinfection efficacy. The lamps operate at 115 v (39 w) and the UVBot can carry a maximum of 4 lamps. By default, we connected two lamps in parallel and powered them using the Powkey 200 w rechargeable power supply (146 Wh) that can output 120 v of A/C current. To safely power the lamps, the current is passed through an electrical ballast, a device placed in series with a load to limit the amount of current in an electrical circuit. We chose the Philips Advance Centium ICN-2S54-T because it has a small form factor and connects two lamps together in parallel. These components are summarized in the yellow box of Figure 4.

#### 2.3.2. Autonomous Movement

We used the iRobot Create 2 as the robot platform of our UVBot. The iRobot Create 2 is an affordable STEM resource for educators, students, and developers. It can receive commands and transmit various sensor signals, including IR sensors, bumper sensors, and cliff detection sensors through a serial port. The iRobot Create 2 has integrated cliff detection functions, which prevents the robot from falling downstairs. It also features integrated docking, allowing the robot to automatically dock to its charger when low on power.

All the robotic control computations are performed on a CanaKit Raspberry Pi 3 Model B, which has a 1.4 GHz 64-bit quad-core processor, dual-band wireless LAN, Bluetooth 4.2/BLE, Ethernet, and Power-over-Ethernet support. Its computation resources cover the cost of robot control and localization, map generation, and desktop interface communication. The LiDAR sensor we use is SLAMTEC RPLiDAR A1M8. It has 360-degree scanning, and we set the scanning frequency at 5 Hz. We used it to generate a 2D mapping of the environment that acts as a user interface in our desktop application for the user to view the environment and update it with virtual walls. We communicated with the robot using this desktop application over Bluetooth. We powered the Raspberry Pi 3B and LiDAR using a rechargeable battery pack that was also used to power the lamps, which we discuss below. These components are summarized in the blue box of Figure 4.

### 2.4. Software Development

#### 2.4.1. Function Overview

We developed a Python package that allows the UVBot to accomplish the disinfection task in various indoor environments. The package supports remote control (remote mode) and autonomous disinfection along the walls (wall-following mode). The package also allows for disinfection map generation, editing the resulting map by adding virtual walls, and loading pre-saved maps. Figure 4 shows the diagram of the software architecture. In remote mode, users can manually control the UVBot to move forward, move backward, turn left, and turn right by pressing the corresponding keys in the desktop application. The moving speed of the UVBot in the remote mode is much faster than its disinfection working speed. The remote mode allows users to move the UVBot from one area to another without lifting it. The wall-following mode is the normal disinfecting mode of the UVBot. The robot first moves forward to find a wall. After reaching the wall, the UVBot starts disinfecting along the walls counterclockwise. While in the disinfection process, the LiDAR sensor scans the environment and SLAM generates a map of the working environment for every scan. The SLAM algorithm also combines this map with the current UVBot location. The map marks the walls and obstacles in the working environment. Users can look up the map from the desktop application with an update frequency of 5 Hz. They can also edit the map by adding virtual walls. Virtual walls are added by drawing lines on the map that act as invisible walls. The virtual walls limit the exploration area of the robot, block out unnecessary space, or construct closed disinfection regions for the UVBot to work.

The software runs on three nodes: the communication node, the robot node, and the SLAM node. After the software starts, the communication node continuously searches for the desktop application. When paired, users can choose to load a pre-saved map or directly start the disinfection process. If a pre-saved map is loaded, the UVBot first finds its position in the map, loads the virtual walls, and then switches to the wall-following mode. If users choose not to load a map, the UVBot will treat it as the first run and will switch to the wall-following mode directly. Whether a map is loaded or not, when the robot stops, the user is asked to save the map if desired. While in the wall-following mode, users can look up the most updated map, switch between the remote mode and the wall-following mode, pause, stop, or restart the robot, load another map, or edit the current map through the desktop application. The desktop application queries the most updated map at 5 Hz. Walls and obstacles in the map are presented as black pixels, while the free space is shown in blue. The disinfected areas are highlighted as red lines, and virtual walls are marked with green lines.

#### 2.4.2. Wall Following

We modeled the robot motion kinematics as a Unicycle model. This model approximates a vehicle as a unicycle with a given wheel radius that can spin in place according to a steering angular velocity, *ω*. Since there are 2 active wheels and 1 passive wheel on the Roomba, we can calculate the linear velocity and steering angular velocity *ω* from the distance d between two wheels and the speed $v_{1}$, $v_{2}$ from 2 wheels.
$$v_{uni}=\frac{v_{1}+v_{2}}{2}$$
$$\omega_{uni}=\frac{v_{1}-v_{2}}{d}$$

The motion of the robot functions under six states, which are “find wall”, “reach wall”, “follow wall”, “collision”, “stop”, and “remote”. The commands issued from the desktop are received by the communication node and used to directly make changes to the current robot state. When the software starts, the UVBot loads the “stop” state automatically. The desktop application can turn the robot from the “stop” to the “find wall” state. In the “find wall” state, the UVBot moves forward until it hits an obstacle with its bumper. At that point, the state automatically switches to “reach wall”. Then the robot moves backwards by 3 cm and spins in place counterclockwise until the IR sensor on the right side of the robot senses the wall. These motions ensure that the robot finds a wall and turns in the correct direction for the upcoming disinfection process. After the UVBot stops spinning, the state changes to “follow wall”.

The wall following is the actual disinfection process. The motion of the UVBot for moving along the wall is controlled at two levels. At the first level, we established a closed-loop control using the IR sensor on the right side of the robot to keep the robot 3 cm from the wall as the robot moves along it. The second level is built to maintain the wall-following motion when the first-level control fails. The closed-loop IR sensor-based control can fail when (1) the IR sensor reading is abnormal due to the wall material, for example, a specular surface or a glass wall; (2) the IR sensor has no reading, but the robot cannot pass, for example, a gap smaller than the width of the robot; or (3) the robot meets an obstacle that cannot be avoided with its control dynamic. The second-level control uses the bumper as feedback. When a positive reading is received from the bumper, the UVBot moves back by 3 cm and changes the state from “wall follow” to “collision” temporarily. Then the UVBot turns left by 5 degrees and switches back to the “wall follow” state. This allows the robot to circumvent any obstacles or find a wall to follow. The closed-loop IR sensor-based controller is built to keep the robot-to-wall distance *d* and the forward moving speed *v*. Based on the inverse relationship between the distance and the reading of the IR sensor, we define a gain (*p*) as:$$p=a\left(d-c\left(1+r\right)\right)$$
where *c* is a constant that depends on the property of the IR sensor. In our controller, we set c=4096. r is the IR sensor reading. a=2.5 is an amplifier of the gain. To avoid an extremely large value of *p* caused by tiny sensor reading, we crop the gain as
k=min(max(p, −δ), δ)
where *δ* is the cropping gain, which equals 0.5 in our implementation. The speed of the left and right wheels ($v_{l}$ and $v_{r}$) of the robot are controlled by
$$v_{l}=v\left(1+k\right)$$
$$v_{r}=v\left(1-k\right)$$

#### 2.4.3. Simultaneous Localization and Mapping (SLAM)

Simultaneous localization and mapping (SLAM) is the computational problem of constructing or updating a map for an unknown environment while simultaneously keeping track of the location of the agent. For example, a robot equipped with a LiDAR is moving in an indoor environment while the LiDAR scan and its odometry are recorded. The robot can determine its movement by the odometry alone, but odometry data tend to be unreliable over a long period due to the accumulation of error. The algorithm must compare LiDAR scans in different readings and determine the current position based on the change in the LiDAR scans and odometry. SLAM algorithms are mostly computationally heavy. However, due to the cost limitation and load capacity of the UVBot, we developed a simple but efficient SLAM algorithm to be deployed on mobile platforms using a single-chip machine. To generate a disinfection map in the user interface, we developed our SLAM algorithm based on BreezySLAM [32].

BreezySLAM is a Python implementation of the work tinySLAM [33]. The tinySLAM consists of two main operations: distance calculation and the update of the map. As the SLAM process starts (*t* = 0), it samples N possible robot configurations as beliefs
$$X_{t}=\left\{\begin{array}{cccc} x_{t}^{1}, & x_{t}^{2}, & \cdots , & x_{t}^{N} \end{array}\right\}$$
while the robot moves from time t to time t + 1, the algorithm estimates N new configurations based on the beliefs at time t and the odometry between time t and t + 1. The estimated new configurations are denoted as
$${\bar{X}}_{t}=\left\{\begin{array}{cccc} {\bar{x}}_{t}^{1}, & {\bar{x}}_{t}^{2}, & \cdots , & {\bar{x}}_{t}^{N} \end{array}\right\}$$

For each estimated configuration, the algorithm calculates the likelihood of the configuration as
$$P_{t}=\left\{\begin{array}{cccc} p_{t}^{1}, & p_{t}^{2}, & \cdots , & p_{t}^{N} \end{array}\right\}$$
based on the distance between the current map and the estimated LiDAR scan as if the robot is at the estimated configuration. Then, the algorithm draws N beliefs ($X_{t+1}$) for time t+1 from ${\bar{X}}_{t}$ with the probability $P_{t}$ for each configuration. The map at time t+1 is updated from time t with the new scan by the probability $P_{t}$ for each configuration in $X_{t+1}$. As the algorithm runs, the beliefs converge to the true configuration of the robot.

As the robot moves in the environment, new scans update the map. In the SLAM algorithm, the previous map and the new scan are weighted with contributions of 99.2% and 0.8%, respectively, while updating the map. The map is sent to the mobile user interface via Bluetooth in real time (at a 5 Hz refresh rate). Users can further determine virtual walls and cleaning areas and check the map in the mobile interface.

#### 2.4.4. Localization and Virtual Wall

After a disinfection cycle, the map is stored as an image in the desktop application for future use. The map can also be edited as needed and is transferred as a byte buffer between the robot and the desktop application via Bluetooth. The loaded map is the initial map for the particle filter. As the robot moves, it will locate the actual configuration on the map. The map starts to update once the configuration beliefs converge.

We also developed a provision to create a virtual wall to contain the robot in a region of interest. Users can define a wall by selecting two edge points, $P_{1}$ and $P_{2},$ on the map. The wall edge points are sent as a byte buffer from the desktop application to the UVBot. After the robot receives the edge points, the main loop resumes, and in each loop, the robot-to-wall distance (D) is computed. The perpendicular point $P_{3}$ is also computed and determined if it is between the edge points or on an extension of the wall. As shown in Figure 5, when the robot is away from the wall, the driving strategy remains the same as the wall-following strategy, as described in the previous sections. When D<400 mm and $P_{3}$ is between points $P_{1}$ and $P_{2}$, the robot enters the virtual wall-following algorithm. When the robot enters the 400 mm distance region, it stops moving forward and rotates left until its orientation is parallel to the virtual wall. Since the normal wall following is determined by the IR sensor reading, and the control effort is derived from the right IR sensor, the virtual wall following sticks to the same principle to keep the performance. As the robot starts moving, a virtual value is assigned to the right IR sensor recordings to simulate the distance from the virtual wall. The smaller the distance is, the larger the IR sensor reading is. This is used to mimic the virtual wall. Once the robot moves beyond the virtual wall section, either D>400mm or $P_{3}$ is no longer between points $P_{1}$ and $P_{2}$, the normal wall-following algorithm takes over the control, and the normal driving strategy resumes.

The virtual wall function can also be used to construct a closed-loop trajectory for the UVBot. The UVBot shuts down the disinfection task only when the stop button is manually pressed on the robot or the stop command is given from the desktop application. By adding virtual walls, users can build a complicated trajectory for the robot to follow, such as an interior of a manufacturing plant, laboratory, or an office with desks in many different configurations. We chose to use the virtual walls to ensure coverage of all the interest areas instead of random exploration or pre-set trajectory following. Because the UVBot only has obstacle detection ability at the bottom of the robot (through IR sensors), objects such as desks cannot be sensed. In such situations, a virtual wall comes in handy as a way of pre-determining the path in a complex unstructured environment.

#### 2.4.5. Reducing Human Exposure to UV Radiation

UV irradiation can increase the risk of skin cancer when the skin is directly exposed to UV at a high dose. To reduce the cost of the robot, this version of the UVBot does not include a human body detector to switch the lights on or off depending on the presence or absence of a human. This is why we suggest that the robot should be used only in a closed and empty space. This requirement will not hurdle the effectiveness or performance of the UVBot. The many features of the UVBot discussed above help it operate in an empty space without risking human exposure to UV light, including autonomous navigation and a desktop application to control the UVBot from a safe distance. We leveraged these features to ensure that our experiments were conducted safely.

## 3. Results

### 3.1. Virus Inactivation by 254 nm UV Irradiation

To test the effectiveness of UV irradiation on virus disinfection, we measured the UV intensity at different distances between the UV source and the targeted surface and the effectiveness of the exposure time on virus inactivation at different UV doses.

First, we measured the UV intensity (mW/cm^2^) at distances between 4 and 44 cm from the lamp (Figure 6 and Appendix A
Table A2). High UV attenuation was noticeable at 44 cm (0.64 mJ/cm^2^). Compared to 4 cm, the UV irradiation lost over 90% of its intensity at 44 cm. At 40 cm, the UV preserved around 15% of the intensity (1.13 mJ/cm^2^) compared to a 4 cm distance. These data suggest that a high disinfection rate was expected to be at a 4 to 40 cm distance from the robot. Increasing the distance from the robot will require a longer time of exposure. Based on these data, all the disinfection experiments were conducted at a range of 10 to 40 cm from the UV lamp.

Second, to determine the disinfection ability of the UV robot, we exposed TV to the UV irradiation at different times (10, 20, 30, and 40 s), and at different distances from the UV lamp (10, 20, 30, and 40 cm). The surviving TV was quantified using standard plaque assay and expressed as PFU/mL. The log reduction was calculated as log_10_ (N_0_/N). Figure 7a shows the log reduction as a function of the UV dose. Linear regression was used to describe the correlation between the log reduction and the UV dose. The slopes of these linear regression lines are the inactivation rate constants. Their values are 0.028, 0.053, 0.084, and 0.087 at 10, 20, 30, and 40 cm, respectively. A higher inactivation rate constant means faster inactivation. Because the inactivation rate constants decreased when the distance increased from 10 cm to 40 cm, inactivation became slower at a longer distance. Except for a 50 cm distance, the log reduction values were up to 4, depending on the exposure time. At a 50 cm distance, the intensity was very low, and after 40 s, only one log reduction was observed. We also plotted all data collected for 10, 20, 30, and 40 cm as a function of the UV dose (Figure 7b). These data can be described by linear regression with an R^2^ value of 0.92. This means the virus inactivation depends on the UV dose. In summary, to achieve a 3-log reduction of virus infectivity, the distance should be smaller than 40 cm, or the UV dose should be over 45 mJ/cm^2^.

Third, we determined the exposure time needed for a 3-log reduction at different distances from the UV source (10, 20, 30, and 40 cm). After the virus was deposited on the drywall, the droplets were exposed to the UV for 10, 20, 30, and 40 s. The log reduction was calculated as log_10_ (N_0_/N) and plotted as a function of time in Figure 8a. The kinetics were described with linear regressions, with an R^2^ value of at least 0.96. The inactivation rate constants, which were taken as the slopes of these linear regressions, were 0.28, 0.053, 0.069, and 0.074) for 10, 20, 30, and 40 cm, respectively. While the log reduction reached 4 after 25 to 30 s for distances of up to 40 cm, the log reduction for a 50 cm distance was only 1 log at the highest exposure time of 40 s. We also plotted all data collected for distances of 10, 20, 30, 40, and 50 cm as a function of the exposure time. Again, these data are described as a linear regression with an R^2^ value of 0.97. As shown in Figure 8b, to achieve 3-log viral reduction, at least 30 s of exposure time was required at distances up to 40 cm.

### 3.2. UVBot Work Speed Calculation

The inactivation data of the Tulane virus show that the minimal required dose to achieve a 3-log reduction or 99.9% disinfection was 45 mJ/cm^2^ combined with a minimal disinfection time of 30 s. Since the effective distance of the UVBot ranged between 4 and 40 cm and the distance between the two UV lamps on the UVBot was 24 cm (Figure 9), the effective spectrum of the UVBot is 104 cm. To ensure that each point within the target spectrum reaches the 30 s exposure time, the speed of the UVBot can be calculated as follows:$$L_{e}/T=104\;\mathrm{cm}/30\;s=3.46\;\mathrm{cm}/s$$
where $L_{e}$ is the effective range of the UV spectrum and T is the required disinfection time. Hence, we set the default working speed (or wall-following speed) as 3.15 cm/s to ensure 99.9% virus disinfection.

### 3.3. UVBot Prototype and Autonomous Disinfection Demonstration

The weight of the UVBot is less than 5 kg. The battery life is around 1.9 h and fully charges in 6 h. The prototype of the UVBot is shown in Figure 10. The detailed mechanical design can be found in Section 2.2.

To demonstrate the autonomous disinfection process, we tested the UVBot in two indoor environments: a corridor and an office, which cover most of the use cases. We tested the robot at the entrance of the Health Care Engineering Systems Center of the University of Illinois at Urbana-Champaign. The corridor is 12.5 m × 5 m, with office rooms on both long edges and office desk cells in the middle. The test required the UVBot to disinfect the walls, office windows, office doors, and door handles. We blocked both short edges of the corridor with white cardboard to construct a small loop for the UVBot. A photo of the corridor and the disinfection process maps are shown in Figure 11.

The overall disinfection took 1119 s, with a total travel length of 35 m. The average disinfection speed was 3.13 cm/s, which meets the speed requirement we calculated from the virus experiment. In the wall-following process, the bumper-based control was involved a few times when the UVBot tried to pass a trash bin, showing that our wall-following algorithm is reliable. The performance of the wall-following algorithm can also be indirectly observed from the actual working speed because the actual speed was close to our default wall-following speed (3.15 cm/s) and the bumper-based control significantly slowed the UVBot. The robot trajectory in the map formed a perfect loop, which demonstrates the superior behavior of the SLAM algorithm.

The office environment was a small office cubicle, as shown in Figure 12. After the disinfection process started, we saved the map, as shown in the bottom-left subplot of Figure 12. Next, we loaded the map and added a virtual wall to prevent the UVBot from moving under the desk. The virtual wall is shown in green lines in the top-right and top-bottom subplots of Figure 12. When the UVBot moved into the 400 mm region of the virtual wall, instead of following the real walls (black pixels in the subplots), it followed the custom virtual wall. The disinfection test results for the office cell show the saved map, loaded map, and edited map functions, as well as the performance of the UVBot following a virtual wall. The time to disinfect the office cell was 120 s, with a travel distance of 3.6 m. The average working speed of 3 cm/s was a bit slower than the default working speed because of the delay caused by the spinning time in corners.

## 4. Discussion

The burden of noroviruses in the field of occupational health has gained significant attention [8,12,34,35]. It has been shown that the number of staff and workers who are infected during NoV outbreaks is higher than the number of infected patients and customers. This leads to a significant economic loss in work time and health expenses, which makes the prevention of virus spread imperative. Herein, we provided a highly efficient, low-cost UVBot that can disinfect various types of workspaces at a sufficient speed, with low labor requirements and no chemical residuals. Due to the lack of a robust cell line to grow NoVs, we used TV as a surrogate for human norovirus to evaluate the UVBot’s effectiveness. TV shares many similarities with human noroviruses, such as the genome and capsid structure, and utilizes the same receptor that NoV uses to attach to the host cell. Therefore, TV is widely used as a surrogate for human noroviruses to study different disinfection approaches [15,16,36]. The data of this work show that UV irradiation was able to achieve a 3-log reduction (99.9 % disinfection) of TV within 30 s when the UV dose was at 45 mJ/cm^2^. The data agree with our previous study showing that over a 4.5-log reduction of TV was achieved at a UV dose of 30 mJ/cm^2^ in wastewater and that the UV was able to denature the TV capsid protein and damage the virus genome [17]. The UVBot was able to deliver this high UV dose within 104 cm of the UVBot. This enables the UVBot to disinfect surfaces and the air between the robot and the target surface. This high dose is expected to inactivate many other airborne viruses and bacteria. For instance, studies have shown that enveloped viruses, such as SARS-CoV-2, which causes COVID-19, and influenza virus are more susceptible to UV irradiation than non-enveloped viruses. For instance, Biasin et al. (2021) reported that 3.7 mJ/cm^2^ of UV-C was enough to eliminate 3 logs of SARS-CoV-2 in culture medium [37]. A 99.9% reduction of aerosolized influenza virus (H1N1) was reported at 15 mJ/cm^2^ at relative humidity ranging from 25 to 84% [38]. Adenovirus type 2 is highly resistant to UV irradiation; however, a UV dose of 30 mJ/cm^2^ at 254 nm UV irradiation reduced the adenovirus infectivity by 2 logs (99% disinfection) [30]. Moreover, Zhang et al. (2019) illustrated that Gram-positive antibiotic-resistant bacteria were decreased in wastewater effluent after UV treatment at a UV dose of 10 mJ/cm^2^ or higher. The study also reported that the horizontal gene transfer of antibiotic-resistance genes among the treated bacteria was reduced by 6–15% [39]. Taken together, the UV dose that the UVBot provides (45 mJ/cm^2^) is enough to inactivate most of the known airborne and enteric viruses and bacteria, and it can efficiently reduce the spread of these pathogens in occupational spaces.

The low-cost design of the UVBot allows this type of disinfection protocol to be much more accessible. The UVBot costs less than USD 1000, while still being lightweight and having autonomous navigation capabilities. Its custom-developed navigation software and user-friendly desktop application allow users to control the robot remotely and ensure the disinfection of the targeted area. The software also allows the user to visualize the robot’s path after the disinfection route has been completed. This is accomplished using a 2D LiDAR to allow for a simultaneous localization and mapping (SLAM) algorithm to draw a map of the space being disinfected. These features make the UVBot a practical tool to disinfect most occupational spaces, such as classrooms, patient rooms, corridors, and shops. To further ensure its availability, we are making all aspects of the design available to any organization by simply filling out a form (https://healtheng.illinois.edu/about/researchareas/robotics/uvbot (accessed on 15 September 2022). We also have our software available on GitHub (https://github.com/wangfanxin/UVBot accessed on 15 September 2022). Other organizations can build and improve on the design. To test the reproducibility of our design, we collaborated with the Whiteside Area Career Center (WACC) to help them build a UVBot for them to use in their facilities. A team of high school students was able to reproduce our design with minimal guidance from our team, showing the ease with which the UVBot can be implemented into a facility.

We also plan to improve the design of the UVBot. First, 254 nm UV light exposure is dangerous for humans [40]. To ensure the safe operation of the UVBot, we recommend operating it in the absence of humans. However, to further ensure that the robot is safe, computer vision techniques will be used to determine if humans are present to shut off the lights. Second, a more robust path-planning solution will be implemented to optimize surface disinfection that considers the 3D environment. In the current design, the robot autonomously navigates environments using a simple wall-following algorithm, along with user input in the form of virtual walls. In future works, we plan to first map the environment as a 3D point cloud and then use these data to determine the optimal path. Furthermore, many targeted environments are dynamic in nature. Obstacles may be placed in front of the UVBot that were not there during the planning phase. Robust collision avoidance will be needed to circumvent this challenge. Finally, we plan to optimize the time taken to disinfect large surfaces by allowing these robots to work in tandem, either from a centralized server or through robot-to-robot communication.

## 5. Future Work

The current design and software support the UVBot to perform basic autonomous disinfection with 99.9% virus inactivation in various indoor environments. Users can interact with the UVBot through a desktop application. However, there are still several issues and improvements we are working on.

Collision detection on the top part of the UVBot: The UVBot collects environment information from IR sensors and bumpers at the bottom and from the LiDAR on the top. The UVBot cannot provide collision avoidance when it encounters obstacles that are above the robot’s base (tabletop for example). We plan on adding sensors surrounding the UV lights and integrating them with the UVBot control, such that the UVBot can avoid collisions at various heights to ensure safety and robustness.

Autonomous human detection: The UV light at 254 nm irradiation used here is harmful to humans, so we are planning to add a feature that will temporarily shut down the UV light when a human is sensed within the work environment of the robot.

Improved docking: In the current software design, the UVBot can disinfect for about 240 min after one full charge. Some large indoor environments will need multiple charges to cover the entire area. We are adding a new feature that will allow the UVBot to continue unfinished disinfection tasks automatically after charging is complete.

Multi-robot working in a network: Some large indoor environments will need multiple UVBots to work at the same time for efficiency. To ensure collaboration and information sharing between these UVBots, we are planning on adding a new mode to allow multiple UVBots to work in a network or as a swarm. In this new mode, working zones for each UVBot could be assigned automatically or by users on a shared map. Reinforcement learning could be used in this situation for increasing the effectiveness of the overall sanitizing [41], or users can set up schedules for each UVBot to optimize their working and battery charging times.

## Figures and Tables

**Figure 1 sensors-22-08926-f001:**
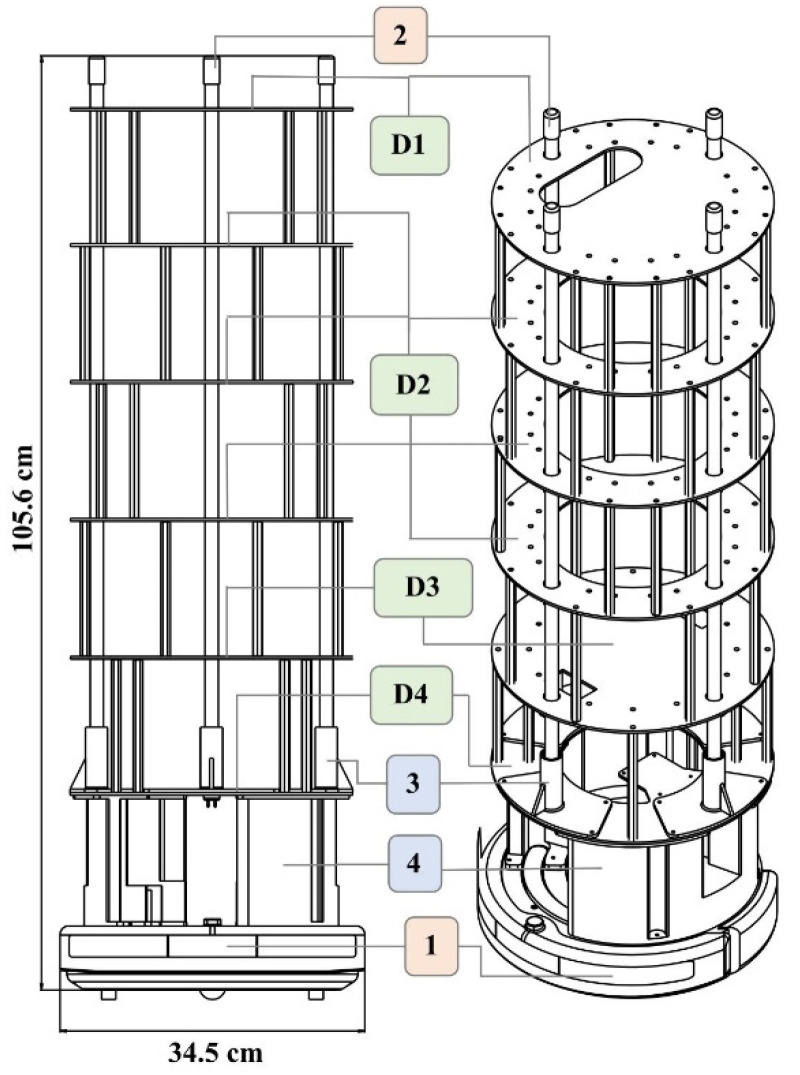
UVBot overall assembly and dimensions. (1) iCreate2 mobile robot, (2) lamps, (3) lamp holder, and (4) base structure. (D1) to (D4) are laser-cut support structures. All components were either purchased off the shelf (salmon-colored boxes), 3D printed in PLA using FDM printing (blue-colored boxes), or laser-cut (green-colored boxes).

**Figure 2 sensors-22-08926-f002:**
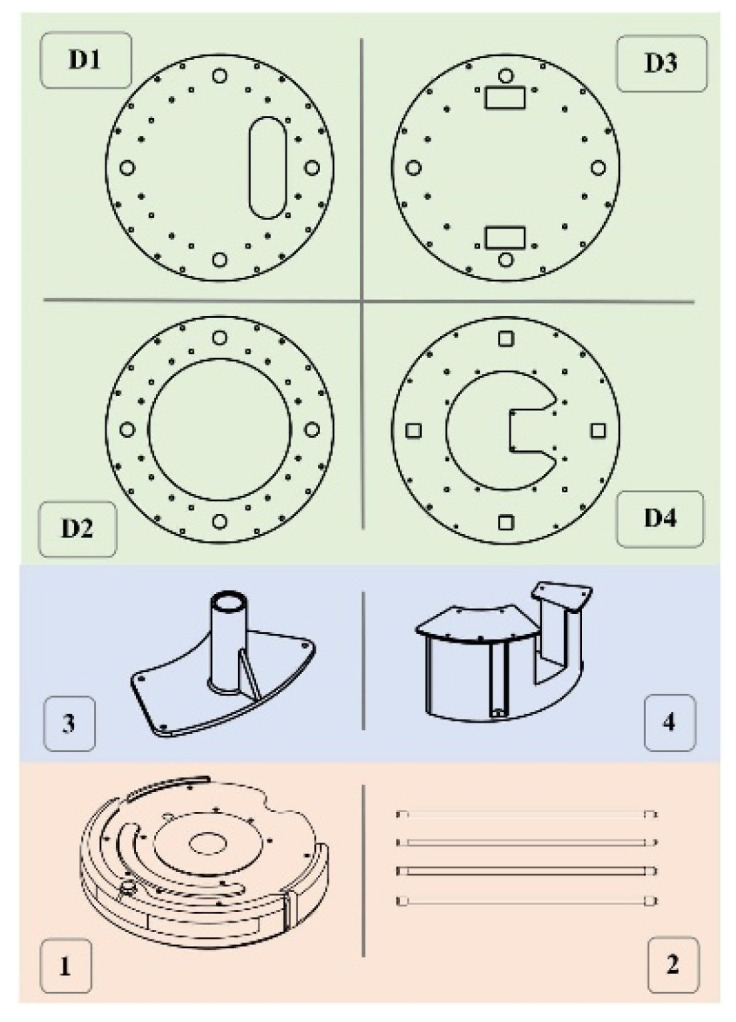
UVBot key mechanical components. (D1) Designed to mount the LiDAR, (D2) serves as support, (D3) used to house electronic components such as power and the ballast, and (D4) serves to mount the Raspberry Pi 3B. (1) is the iRobot Create 2, (2) is the UV lamps, (3) is 3D printed lamp holder, (4) is 3D printed support structure. All components were either purchased off the shelf (salmon-colored boxes), 3D printed in PLA using FDM printing (blue-colored boxes), or laser-cut (green-colored boxes).

**Figure 3 sensors-22-08926-f003:**
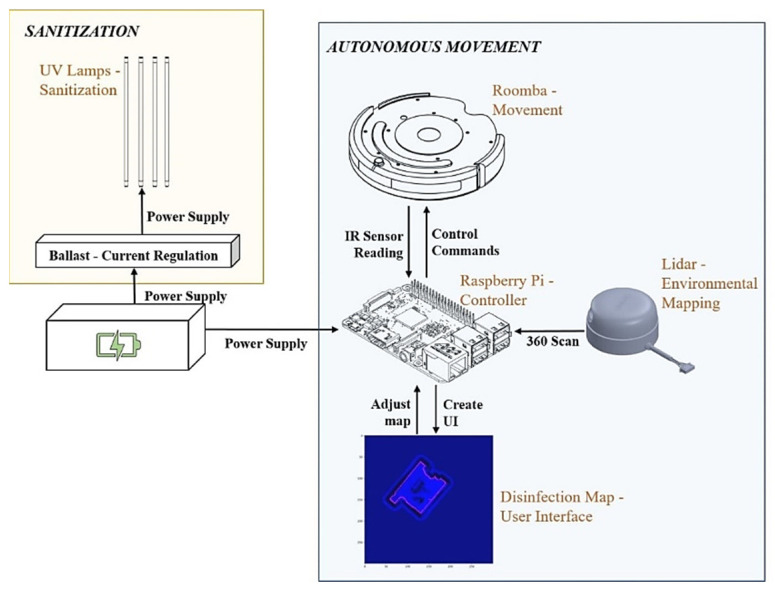
UVBot electrical design overview. In orange is the sanitization module containing the UV lamps and the ballast. In blue is the autonomous movement module containing the mobile robot, the LiDAR sensor, and the microcontroller. Both modules are powered by the same power supply.

**Figure 4 sensors-22-08926-f004:**
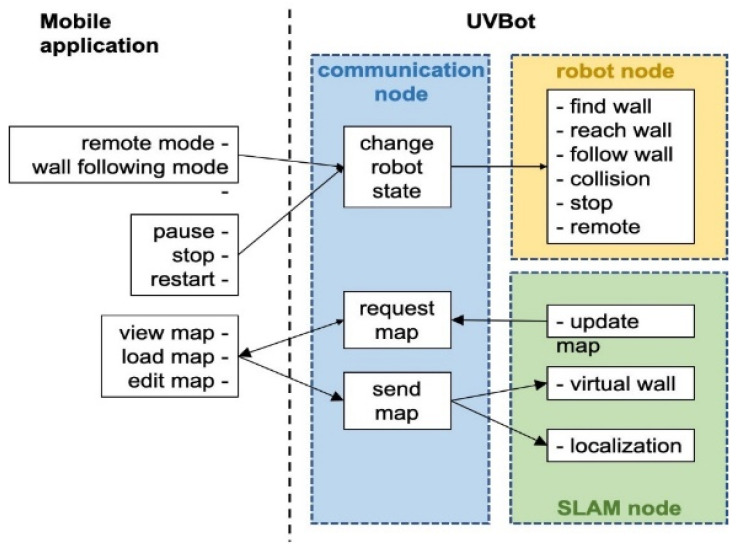
Diagram of the software architecture.

**Figure 5 sensors-22-08926-f005:**
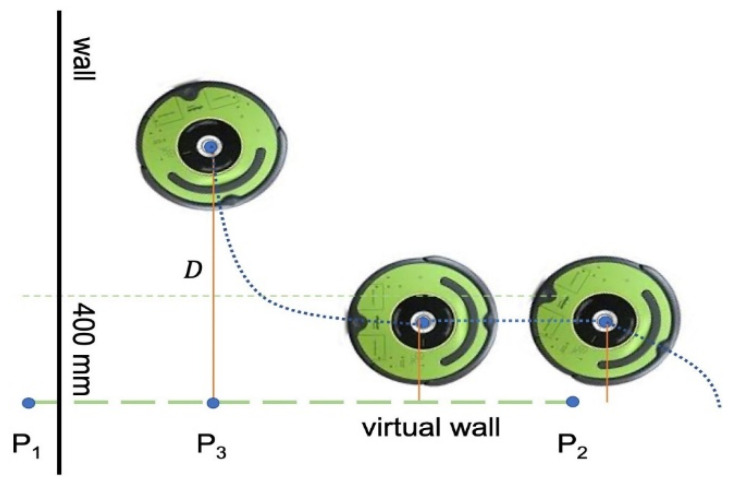
Virtual wall function.

**Figure 6 sensors-22-08926-f006:**
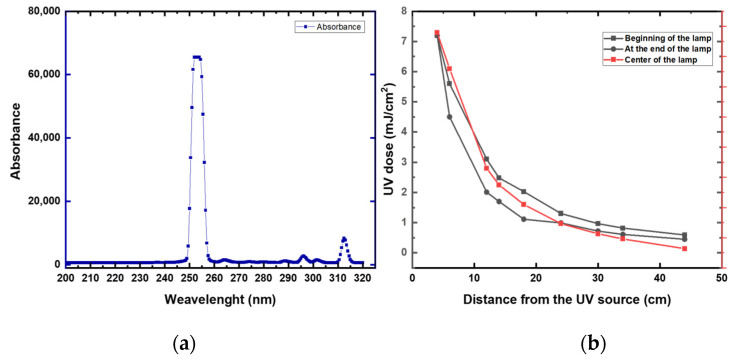
(**a**) UV intensity at different distances from the UV lamp and at different locations of the UV lamp. (**b**) UV spectrum of the studied UV lamp.

**Figure 7 sensors-22-08926-f007:**
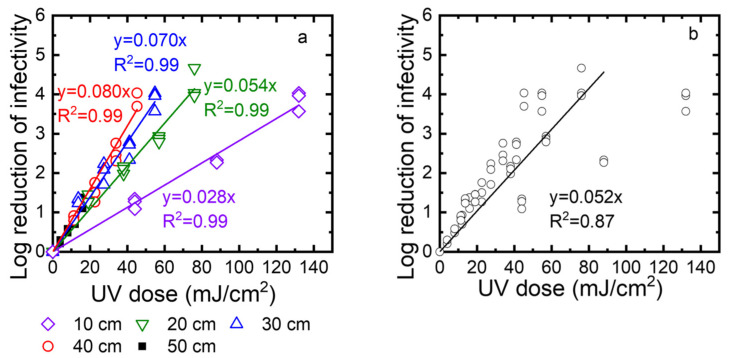
(**a**) Effect of UV dose and distance on the virus inactivation. (**b**) TV inactivation as a function of the UV dose for distances of 10, 20, 30, and 40 cm. Note that the regression does not include the points for the highest UV dose obtained when the distance was 10 cm. Log reduction of infectivity is log_10_ (N_0_/N), where N_0_ is the infectivity of the viral suspension before exposure to UV, and N is the infectivity of this suspension after UV exposure.

**Figure 8 sensors-22-08926-f008:**
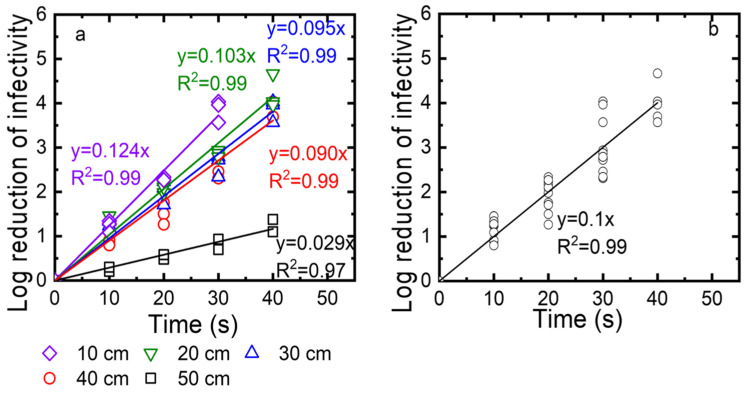
(**a**) TV inactivation as a function of time measured for different distances (10, 20, 30, 40 cm). (**b**) Linear regression for TV inactivation as a function of time measured for distances of 10, 20, 30, and 40 cm. Log reduction of infectivity is log_10_ (N0/N), with N0 as the infectivity of the viral suspension before exposure to UV and N as the infectivity of this suspension after UV exposure.

**Figure 9 sensors-22-08926-f009:**
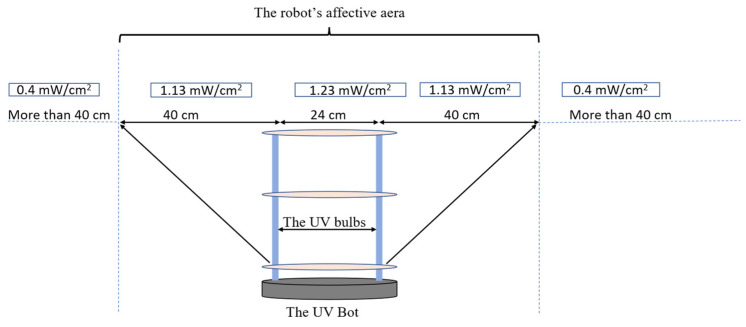
The UVBot effective distance and targeted disinfection area.

**Figure 10 sensors-22-08926-f010:**
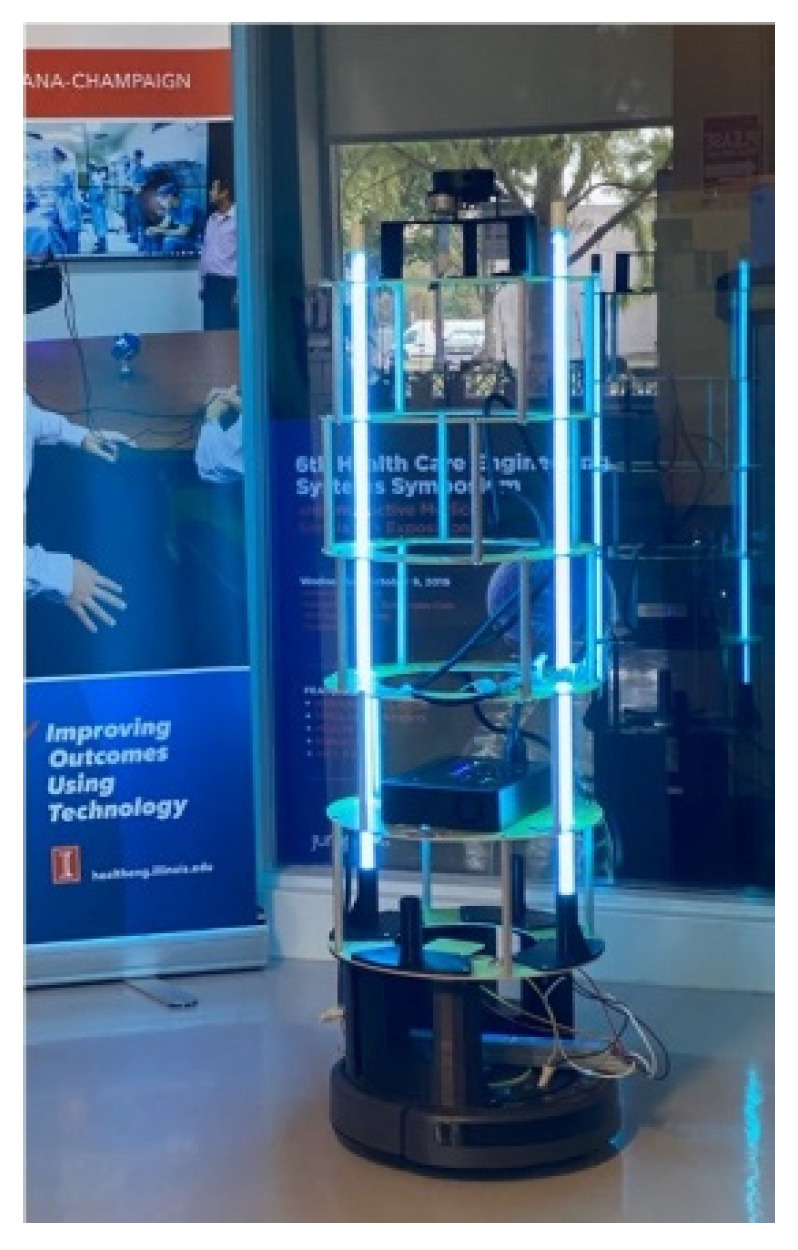
Functional prototype of the UVBot.

**Figure 11 sensors-22-08926-f011:**
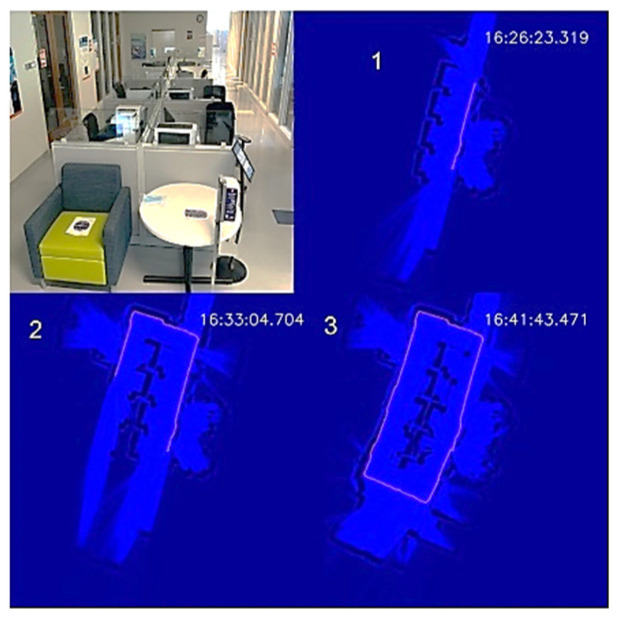
UVBot disinfection test in a corridor. (1–3) Progression of the UVBot in this environment.

**Figure 12 sensors-22-08926-f012:**
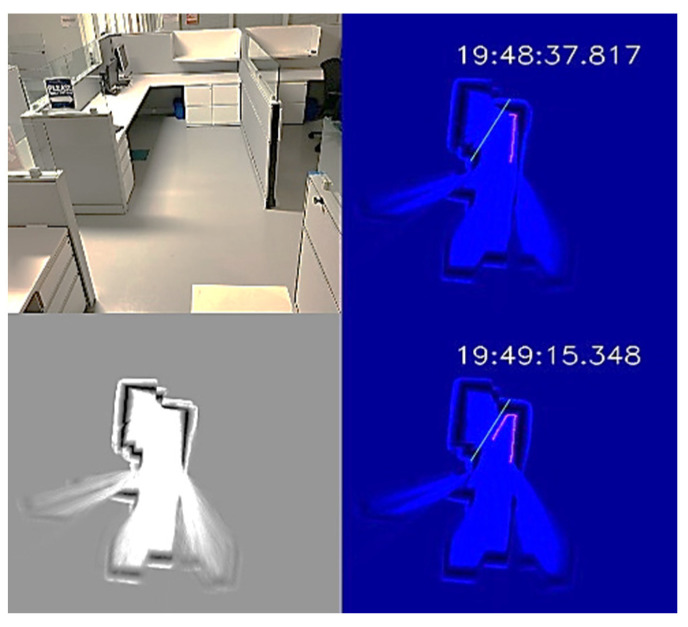
UVBot disinfection test in an office cell using a virtual wall. **Top-left**: Photo of the office cell; **bottom-left**: the stored map; **top-right** and **bottom-right**: UVBot disinfection map with the virtual wall.

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
