# Peer review of "Low-Cost UVBot Using SLAM to Mitigate the Spread of Noroviruses in Occupational Spaces"

_sensors, 2022, doi:10.3390/s22228926_

Round 1

Reviewer 1 Report

This paper develops a low-cost UVBot using SLAM to perform disinfection in the occupational spaces. The robot uses a mobile robot integrated with the UV-light to achieve the autonomous disinfection for indoor environments. The advantage is its low-cost material and parts. Validations are done to show its effectiveness and capabilities. Some minor comments can be found below.

1. Does the robot only for norovirus? Can it be used to other viruses, like COVID-19? The generalization of the developed robotic system can be illustrated.

2. The UV irradiation is provided by the long UV lamps. What is the potential disinfection area? What is the safety distance between the robot and human? They could be added in the text.

3. Some well-done techniques, like SLAM, is used for this robot. Suggest reducing the contents of this part. Try to highlight what are the newly-added functions.

4. What is the disinfection time for the UV-light? How to evaluate when the task is finished? They could be claimed in the text.

5. Suggest enlarging Fig. 10, and add more details, because it is the first time to show the real prototype in the manuscript.

6. The text in Fig. 6: Wave length -> Wavelength?

Author Response

Dear reviewer:

Thank you for your kind comments, here are our point-by-point responses:

1. Does the robot only for norovirus? Can it be used to other viruses, like COVID-19? The generalization of the developed robotic system can be illustrated.

Response1:

The main target of the UVBOT was to reduce the norovirus burden (norovirus is a non-enveloped virus), but the UV-irradiation can inactivate the enveloped viruses such as SARS-CoV-2, that causes COVID-19, easer. Most studies showed that a few seconds of UV-irradiation at 3 mJ/cm2will be more than enough to inactivate 99.9% SARS-CoV-2 on surfaces. That makes this UVBOT able to inactivate wide range of pathogenic viruses including the SARS-CoV-2.

2. The UV irradiation is provided by the long UV lamps. What is the potential disinfection area? What is the safety distance between the robot and human? They could be added in the text.

Response2:

The potential disinfection area is mentioned in section 3.2. We added a summary in the abstract part to make it clearer.

3. Some well-done techniques, like SLAM, is used for this robot. Suggest reducing the contents of this part. Try to highlight what are the newly-added functions.

Response3:

We think the content is needed for complete illustration of the functionality of the robot. We further introduce our software architecture and virtual wall behavior for better understanding of the robot.

4. What is the disinfection time for the UV-light? How to evaluate when the task is finished? They could be claimed in the text.

Response4:

The disinfection time and evaluation are well stated the result section.

‘’The inactivation data of Tulane virus showed that the minimal required dose to achieve 3-log reduction or 99.9% disinfection was 45 mJ/cm2 combined with the minimal disinfection time of 30 s.’’

We further calculated the relative speed of the UVBot to carry out this disinfection rate at 3.5cm/s.

5. Suggest enlarging Fig. 10, and add more details, because it is the first time to show the real prototype in the manuscript.

Response5:

We added to the new script with more description.

6. The text in Fig. 6: Wave length -> Wavelength?

Response6:

Yes. We have changed the typo

Reviewer 2 Report

This manuscript presents Low-cost UVBot using SLAM to mitigate the spread of noroviruses in occupational spaces. The first claim is that this robot is low-cost ($1000) compared to commercial robots ($35.000).

However, it is common for lab prototype robots to be that low-cost, and even some robots are cheaper than $1000. The question is whether the so-claim low-cost robot can be as effective as the high-cost commercial robots. Furthermore, applying Lidar for SLAM is a common practice in self-navigating robots, moreover a wall following robot. Hence, this manuscript lacks novelty and contribution. 

The objective of this UV-light-based disinfectant robot is very similar to the current covid-19 robot application.

The authors do not give the kinematics and dynamic analysis of the trajectory generation for this robot. Since this robot is meant to be implemented among people, it is very important to prove that this robot is stable by presenting stability analyses such as the Lyapunov stability analysis. The stability analysis can be related to subsection 3.2. UVBot Work Speed Calculation.

Since the Journal to publish this manuscript is "Sensor," the authors should also emphasize analyzing the sensors installed on the robot more.

Author Response

Dear reviewer:

Thank you for your kind comments, here are our point-by-point responses:

1. This manuscript presents Low-cost UVBot using SLAM to mitigate the spread of noroviruses in occupational spaces. The first claim is that this robot is low-cost ($1000) compared to commercial robots ($35.000).

However, it is common for lab prototype robots to be that low-cost, and even some robots are cheaper than $1000. The question is whether the so-claim low-cost robot can be as effective as the high-cost commercial robots. Furthermore, applying Lidar for SLAM is a common practice in self-navigating robots, moreover a wall following robot. Hence, this manuscript lacks novelty and contribution. 

Response1:

The objective of this UV-light-based disinfectant robot is very similar to the current covid-19 robot application. We also had a local high school reproduce the design of the robot and are allowing them to utilize the design to disinfect their classrooms.

2.The authors do not give the kinematics and dynamic analysis of the trajectory generation for this robot. Since this robot is meant to be implemented among people, it is very important to prove that this robot is stable by presenting stability analyses such as the Lyapunov stability analysis. The stability analysis can be related to subsection 3.2. UVBot Work Speed Calculation.

Response2:

The current model we developed is using the 254 nm UV light bulb, which would cause harm to human, especially to human eyes. Thus, the human coordinating behavior of the robot is not investigated yet. However, we are looking into the new technology of 222 nm UV light bulb which would be human-friendly. Under this condition, human interaction should be taken into account. And this is illustrated in the future work section. We added the kinematics analysis in the main content.

3.Since the Journal to publish this manuscript is "Sensor," the authors should also emphasize analyzing the sensors installed on the robot more.

Response3:

As a comprehensive application, not only sensors but also the whole system is introduced here. Also sensors have been applied in the evaluation section which we measure the effectiveness of UV light in section 3.1.

There are two sensors were used to measure the UV intensity: The UV spectrum was measured using a BLK-C-50 spectrometer (StellarNet Inc., Tampa, FL). The UV intensity was measured using Radiometer/ Photometer (International Light, Model IL1400A) with QNDS2 # 24710 detector.

We have added this content to the main text.

Round 2

Reviewer 2 Report

This paper proposes Low-cost UVBot using SLAM to mitigate the spread of noroviruses in occupational spaces. The claim of low-cost is relative since it is compared to commercial products. The robot function to mitigate the spread of noroviruses is similar with robots applied to beat covid-19 pandemic. The SLAM method is also a common method for navigation. Hence, there is not enough novelty in this research. This paper does not provide kinematics and dynamics modeling and stability analysis, which are features necessary for designing robots implemented in a public area.

Author Response

Dear reviewer,

Thank you for your comments. Here are our responses.

Please note that we did address many of the concerns in our Round 1 response. However, we would like to emphasize that the complete design and manufacturing was done with the cost of the product in mind. We believe that for public health angle this is a truly innovative approach since most systems for such applications are too costly for general use. By integrating advanced sensors that is available for reasonable cost, this paper demonstrates that very cost-effective robotic systems can be developed and implemented effectively. This has an impact around the world where the cost of many commercial products is beyond their reach. Our robotic system can help in making environment safer and help in controlling infections everywhere in the world. In fact, several schools and educational institutions have downloaded the entire design from our Github site. One school shared their project which was a part of their STEM program. They delivered the robot local health care organizations.